# Predictive value of neutrophil to lymphocyte ratio for clinical outcomes in liver cirrhosis: A systematic review and meta-analysis

**Jingjing Lin** [ORCID]*, **Mengna Huang, Lina Shen**

Department of Infectious Diseases and Hepatology, Ningbo Medical Center Lihuili Hospital, Ningbo, Zhejiang, China

* 195554409@qq.com

## Abstract

### Background

The neutrophil-to-lymphocyte ratio (NLR) is a simple yet potent marker that has been established as an independent predictor of mortality in patients with cirrhosis. However, consensus is lacking regarding its prognostic value for predicting mortality risk in these patients. Therefore, we conducted a meta-analysis to clarify its clinical significance.

### Methods

We performed a literature search in PubMed, Web of Science, Cochrane Library, and Embase for studies published from database inception until January 1, 2025. Studies focusing on the diagnosis of cirrhosis were included, while those involving children and neonates were excluded. Odds ratio (OR) and its 95% confidence interval (CI) were calculated using the random-effects model. Sensitivity analysis was conducted to assess heterogeneity, subgroup analysis to explore sources of heterogeneity, and Egger's test to evaluate publication bias. All analyses were performed using Review Manager (v5.4.1) and Stata (v15.0).

### Results

A total of 18 studies involving 7,714 patients were included. Significant associations were obserced between the NLR and mortality (OR = 1.16, 95% CI: 1.10–1.22; P < 0.00001), infection (OR = 0.67, 95% CI: 0.37–0.96; P < 0.0001), and ascites (OR = 2.03, 95% CI: 1.43–2.88; P < 0.0001). However, no signficant correlation was found between the NLR and hepatic encephalopathy (OR = 1.33, 95% CI: 0.89–2.00; P = 0.16). Subgroup analysis indicated that heterogeneity was partly attributable to variations in NLR cut-off values. Egger's test revealed significant publication bias only for mortality.

**Data availability statement:** All relevant data are within the paper and its Supporting information files.

**Funding:** The author(s) received no specific funding for this work.

**Competing interests:** The authors have declared that no competing interests exist.

## Conclusion

NLR is a well-validated prognostic biomarker for mortality, infection, and ascites in patients with cirrhosis.

## Systematic review registration

PROSPERO, identifier CRD420251031417.

## 1. Introduction

Liver cirrhosis, the end stage of chronic liver disease, is characterized by extensive fibrosis, pseudolobule formation, and abnormal vascular proliferation. Patients in the compensated stage are typically asymptomatic, whereas decompensated cirrhosis manifests as portal hypertension and severe hepatic dysfunction. This condition carries a high risk of multi-organ failure and mortality due to complications such as sepsis, ascites, gastrointestinal hemorrhage, hepatorenal syndrome, hepatic encephalopathy, and hepatocellular carcinoma [1]. As a major global health burden, liver disease accounts for over 1 million annual deaths worldwide [2]. In China, which bears one of the highest disease burdens, the prevalence of cirrhosis ranges from 2% to 10% [3]. Approximately 20% of cirrhotic patients progress to decompensation annually, resulting in a markedly reduced 5-year survival rate of 15%−40% [4,5]. Disease severity significantly impacts mortality risk in this population [6,7], with systemic inflammation frequently observed in advanced stages [7,8] and associated with adverse outcomes [9]. Following a diagnosis of cirrhosis,comprehensive management should be initiated promptly, focusing on etiology-specific treatment and complication prevention. For patients with suboptimal response to pharmacotherapy, interventional procedures or liver transplantation may be considered when indicated [1]. Although transplantation remains the definitive treatment for decompensated cirrhosis, its accessibility is limited by donor organ shortages and substantial costs [10,11]. Consequently, early identification of high-risk cirrhotic patients through prognostic stratification is crucial for optimizing transplant allocation. The development of reliable markers correlating with disease severity would facilitate rapid risk stratification, enabling timely therapeutic interventions to improve clinical outcomes.

The progression of liver cirrhosis is aggravated by the synergistic interaction of chronic systemic inflammation and immunodeficiency [12]. Although the Child-Turcotte-Pugh (CTP) and Model for End-Stage Liver Disease (MELD) scoring systems remain the gold standards for prognostic prediction in cirrhosis [13–17], these tools fail to incorporate immune dysfunction – a critical determinant of outcomes in decompensated cirrhosis [18]. The neutrophil-to-lymphocyte ratio (NLR) quantifies the imbalance between innate and adaptive immune responses [19,20]. As neutrophils mediate pro-inflammatory activity and lymphocytes modulate immune regulation [21], the NLR – calculated from routine complete blood count data – serves as an accessible biomarker of systemic inflammation. Substantial evidence correlates elevated NLR values with adverse clinical outcomes, including higher mortality rates,

in cirrhotic patients [22–26]. Notably, Biyik et al. [27] established NLR as a mortality predictor independent of MELD and CTP scores. Complementary findings by Magalhães et al. [28] support the utility of NLR in stratifying infection risk in this patient population.

Despite numerous clinical investigations examining the prognostic utility of the NLR in cirrhosis, a systematic meta-analysis is lacking in the current literature. To address this knowledge gap and evaluate the potential of NLR as a prognostic indicator, we performed a comprehensive systematic review and meta-analysis of contemporary evidence.

## 2. Materials and methods

### 2.1. Literature search strategy

We conducted a systematic literature search across four databases (PubMed, Web of Science, Cochrane Library, and Embase) for studies involving adults diagnosed with liver cirrhosis, covering publications from database inception through January 1, 2025. Our search strategy incorporated the following MeSH terms: "Liver Cirrhosis", "Neutrophils", "Lymphocytes", and "Ratio". To maximize retrieval, we additionally utilized PubMed's 'related articles' feature and performed manual reference screening of included studies. This approach facilitated the identification of both prospective and retrospective studies relevant to our analysis. The complete search syntax is provided in Table 1.

### 2.2. Inclusion and exclusion criteria

Inclusion criteria: (1) Hospitalized adult patients with clinically confirmed liver cirrhosis; (2) Compliance with international diagnostic/management guidelines incorporating NLR as a prognostic marker; (3) Cohort or case-control study designs; (4) Reported hazard ratios (HR) or odds ratios (OR) with 95% confidence intervals (CI) derived from Cox proportional hazards models or Kaplan-Meier analysis.

**Table 1. Literature search strategy.**

**1. Pubmed-264**

(((("Neutrophils"[Mesh]) OR (((((((((((((Neutrophil) OR (Leukocytes, Polymorphonuclear)) OR (Leukocyte, Polymorphonuclear)) OR (Polymorphonuclear Leukocyte)) OR (Polymorphonuclear Leukocytes)) OR (Polymorphonuclear Neutrophils)) OR (Neutrophil, Polymorphonuclear)) OR (Polymorphonuclear Neutrophil)) OR (LE Cells)) OR (Cell, LE)) OR (LE Cell)) OR (Neutrophil Band Cells)) OR (Band Cell, Neutrophil)) OR (Neutrophil Band Cell:))) AND (("Lymphocytes"[Mesh]) OR (((((Lymphocyte) OR (Lymphoid Cells)) OR (Cell, Lymphoid)) OR (Cells, Lymphoid)) OR (Lymphoid Cell)))) AND (ratio)) AND (("Liver Cirrhosis"[Mesh]) OR (((((Hepatic Cirrhosis) OR (Cirrhosis, Hepatic)) OR (Cirrhosis, Liver)) OR (Fibrosis, Liver)) OR (Liver Fibrosis)))

**2. Embase-957**

(neutrophils OR neutrophil OR (leukocytes, AND polymorphonuclear) OR (leukocyte, AND polymorphonuclear) OR (polymorphonuclear AND leukocyte) OR (polymorphonuclear AND leukocytes)OR (polymorphonuclear AND neutrophils) OR (neutrophil, AND polymorphonuclear) OR (polymorphonuclear AND neutrophil) OR (le AND cells)OR (cell, AND le) OR (le AND cell)OR (neutrophil ANDband ANDcells) OR(band ANDcell, ANDneutrophil)OR(neutrophil ANDband ANDcell))AND (lymphocytes OR lymphocyte OR (lymphoid AND cells) OR (cell, AND lymphoid) OR (cells,AND lymphoid) OR (lymphoid AND cell))AND ratio AND (liver AND cirrhosis OR (hepatic ANDcirrhosis) OR (cirrhosis,AND hepatic) OR(cirrhosis,AND liver) OR (fibrosis,ANDliver) OR(liver ANDfibrosis))

**3. Cochrane-15**

((Neutrophils or (Neutrophil or Leukocytes, Polymorphonuclear or Leukocyte, Polymorphonuclear or Polymorphonuclear Leukocyte or Polymorphonuclear Leukocytes or Polymorphonuclear Neutrophils or Neutrophil, Polymorphonuclear or Pooeeed Band Cell:) and (Lymphocytes or (Lymphocyte or Lymphoid Cells or Cell, Lymphoid or Cells, Lymphoid or Lymphoid Cell) and ratio and (Liver Cirrhosis or (Hepatic Cirrhosis or Cirrhosis, Hepatic or Cirrhosis, Liver or Fibrosis, Liver or Liver Fibrosis))).af.

**4. Web of science-268**

((((Neutrophils) OR (((((((((((((Neutrophil) OR (Leukocytes, Polymorphonuclear)) OR (Leukocyte, Polymorphonuclear)) OR (Polymorphonuclear Leukocyte)) OR (Polymorphonuclear Leukocytes)) OR (Polymorphonuclear Neutrophils)) OR (Neutrophil, Polymorphonuclear)) OR (Polymorphonuclear Neutrophil)) OR (LE Cells)) OR (Cell, LE)) OR (LE Cell)) OR (Neutrophil Band Cells)) OR (Band Cell, Neutrophil)) OR (Neutrophil Band Cell:))) AND ((Lymphocytes) OR (((((Lymphocyte) OR (Lymphoid Cells)) OR (Cell, Lymphoid)) OR (Cells, Lymphoid)) OR (Lymphoid Cell)))) AND (ratio)) AND ((Liver Cirrhosis) OR (((((Hepatic Cirrhosis) OR (Cirrhosis, Hepatic)) OR (Cirrhosis, Liver)) OR (Fibrosis, Liver)) OR (Liver Fibrosis))) (Topic)

Exclusion criteria: (1) Neonatal or pediatric cirrhosis studies; (2) Studies reporting only unadjusted ORs without 95% CIs; (3) Studies with insufficient data for extraction. The study selection process rigorously adhered to PRISMA (Preferred Reporting Items for Systematic Reviews and Meta-Analyses) 2020 guidelines [29].

### 2.3. Data extraction and quality assessment

The data extraction protocol systematically captured the following variables from each study: first author and publication year, study period and geographical region, study design,demographic parameters (sex distribution, mean age), predefined NLR cutoff values, mortality rates, infectious complications, ascites development, hepatic encephalopathy episodes, hospital readmissions, hepatocellular carcinoma incidence, and adjusted odds ratios (ORs) with corresponding 95% confidence intervals for prognostic evaluation. Two independent investigators (M.N.H. and L.N.S.) performed all data extraction procedures, with discrepancies resolved through structured discussion. Unresolved conflicts were adjudicated by a senior researcher (J.J.L.). Study quality was assessed using the Newcastle-Ottawa Scale (NOS) [30], with scores ≥6 indicating methodologically robustness.

### 2.4. Statistical method

To determine the prognostic utility of NLR in liver cirrhosis, we performed meta-analyses using pooled odds ratios (ORs) with 95% confidence intervals (95% CIs) under a random-effects modle. (Due to clinical heterogeneity and advance protocol requirements, we continue to use a uniform random-effects model to maintain methodological consistency). Subgroup analyses were conducted to investigate potential sources of heterogeneity, while sensitivity analyses assessed the robustness of the results Publication bias was evaluated through funnel plot symmetry and Egger's linear regression (statistical significance set at $P < 0.05$). All computations were performed using STATA version 15.0 (StataCorp LLC, College Station, TX) and Review Manager 5.4 (The Cochrane Collaboration).

## 3. Results

### 3.1. Identification of relevant studies

Our systematic search identified 1,504 potentially eligible studies across four databases (PubMed, Embase, Cochrane Library, and Web of Science). After duplicate removal and initial screening, 88 articles met preliminary inclusion criteria. Subsequent evaluation excluded 40 studies with incomplete data and 11 unavailable publications, resulting in 37 candidate studies. Further exclusions comprised guidelines or meeting(n = 9), editorials or letters(n = 6), animal experiment(n = 3), and one unavailable full-text publication.. The final meta-analysis incorporated 18 qualifying studies encompassing 7,714 participants [27,28,31–46] (Fig 1).

### 3.2. Study characteristics and quality assessment

The analysis included five prospective [34,36,38,42,44] and thirteen retrospective studies [27,28,31–33,35,37,39–41,43,45,46]. Age distribution revealed four studies focusing exclusively on patients >60 years [32,33,37,39], while fourteen included younger populations (≤60 years) [27,28,31,34–36,38,40–46]. Mortality assessments varied temporally: 90-day (n = 6) [34–36,38,44,45], 28-day (n = 2) [40,43], and 30-day (n = 4) [31,33,39,42]. One study evaluated long-term mortality (12–36 months) [27], while others examined infection risk (n = 2) [28,46], ICU mortality (n = 1) [32], hepatocellular carcinoma incidence (n = 1) [41], and hospital readmissions (30–180 day) (n = 1) [37].

All 18 studies investigated NLR's prognostic value for mortality in cirrhotic adults, employing varying cutoffs: twelve used NLR ≥ 3 [27,28,31–36,38,40,42,44,45], three applied NLR < 3 [41,43,46], and three unspecified thresholds [33,37,39]. Newcastle-Ottawa Scale evaluation demonstrated consistent methodological rigor (all scores >6), indicating robust quality with minimal risk of bias (Table 2).

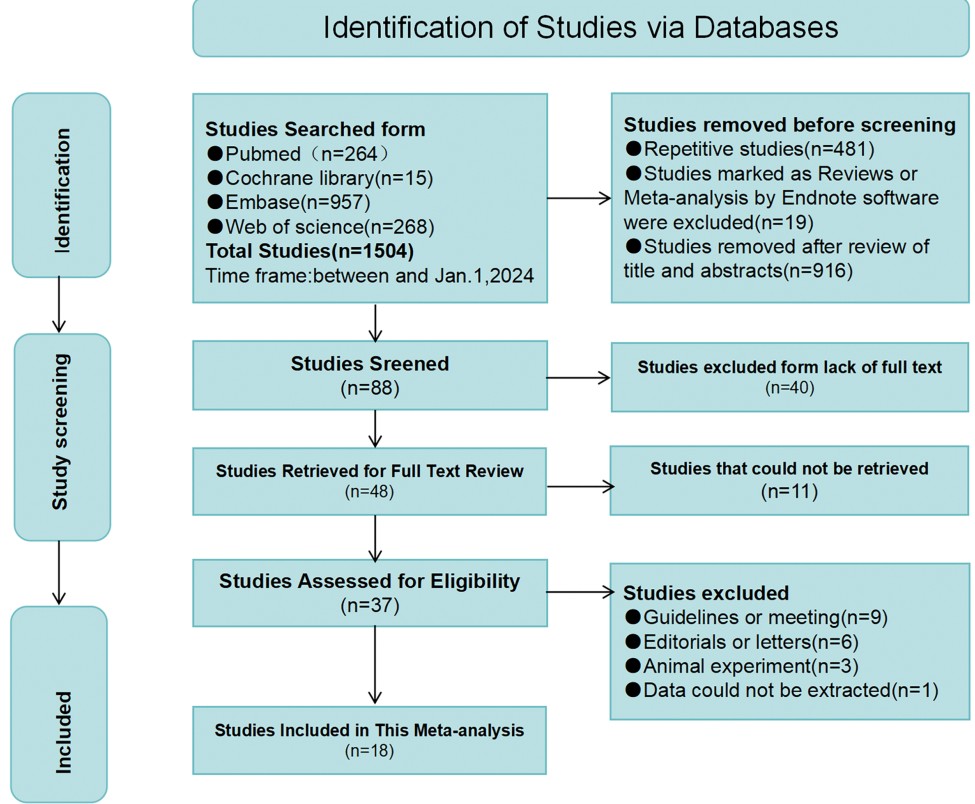

**Fig 1. Flow chart of literature screening.**

### 3.3. Meta-analysis results

**3.3.1. NLR and mortality.** We evaluated the prognostic value of NLR by analyzing twelve cohort studies comprising 5,120 participants. The random-effects model revealed significant inter-study heterogeneity ($I^2$=89%, P<0.00001) and demonstrated a strong association between elevated NLR and mortality in cirrhotic patients (OR = 1.16, 95% CI: 1.10–1.22; P<0.00001; Fig 2A).Stratified analyses by patient age (>60 or ≤60 years), study design (prospective or retrospective), NLR cutoff value (>3 or ≤3), follow-up duration (≥12 or <12 months), and geographic region (Asia vs. Europe) were conducted to explore heterogeneity sources (Table 3). The results showed an association between mortality and follow-up duration (OR = 1.10, 95% CI: 0.93–1.31; P=0.27), and between mortality and average age (OR = 2.37, 95% CI: 0.39–14.36; P=0.35). Heterogeneity was aslo observed with NLR cutoff ($I^2$=47%). NLR demonstrates superior short-term (P<0.00001) versus long-term (P=0.27) mortality prediction in cirrhosis and showed enhanced prognostic utility in patients aged ≤60 years (P<0.00001). The observed heterogeneity primarily stemmed from variability in NLR cutoff thresholds across studies.

We examined the correlation between NLR values and survival status across five cohort studies involving 848 participants. The random-effects model revealed substantial heterogeneity ($I^2$=89%, P<0.00001). Analysis demonstrated significantly higher NLR levels in non-surviving cirrhotic patients compared to survivors (standardized mean difference [SMD] = 1.26, 95% CI: 0.64–1.88; P<0.00001; Fig 2B).

**3.3.2. NLR and Infection.** Six studies provided data on NLR's association with infection risk in liver cirrhosis. Significant heterogeneity was observed ($I^2$=95%, P<0.00001), prompting the use of a random-effects model (Fig 3A). The

**Table 2. Basic characteristics of the included literature.**

| Author | Year | Country | Population | Study design | Male/ Female | Mean age (yrs) | Duration | Patients NO | Outcome Measure | Cut-off value | Quality score | Etiology |
|---|---|---|---|---|---|---|---|---|---|---|---|---|
| Biyik [27] | 2013 | Turkey | cirrhosis | retrospective cohort | 86/59 | 58.9 | 2009.1-2011.12 | 145 | 12-, 24- and 36-month mortality | 4.22 | 7 | Viral (hepatitis B and/or C) /Crypto-genic/Alcoholic/oth-ers rare reasons |
| Chen [38] | 2014 | China | cirrhosis | prospective cohort | 98/29 | 47 | 2011.8-2012.8 | 127 | 90- day mortality | 5 | 9 | NA |
| Kwon [33] | 2015 | Korea | cirrhosis | retrospective cohort | 125/59 | 56.7 | 2011.9-2012.9 | 184 | 30- day mortality | NA | 8 | HBV/HCV/Alc/ others |
| Zhang [31] | 2016 | China | decom-pensated cirrhosis | retrospective cohort | 118/30 | 53.2 | 2014.1-2015.1 | 148 | 30- day mortality | 5 | 7 | HBV |
| Kalra [45] | 2017 | America | cirrhosis | retrospective cohort | 61/46 | 55 | 2002.2-2011.5 | 107 | 90- day mortality | 4 | 7 | Hepatitis C virus/ Alcohol/Hepatitis C virus with alcohol/ Cholestatic/autoim-mune/Nonalcoholic steatohepatitis/ cryptogenic/Other metabolic/genetic |
| Mousa [46] | 2017 | British | decom-pensated cirrhosis | retrospective case-control | 108/72 | 55 | NA | 180 | infection risk | 2.89 | 9 | HBV, HCV |
| Lin [39] | 2018 | China | decom-pensated cirrhosis | retrospective cohort | 204/148 | 60 | 2014.2-2017.2 | 352 | 30- day mortality | NA | 6 | NA |
| Li [40] | 2020 | China | decom-pensated cirrhosis | retrospective cohort | 139/35 | 53.6 | 2017.7-2019.12 | 174 | 28- day mortality | 3.78 | 8 | HBV |
| MACCALi [44] | 2020 | Brasil | cirrhosis | prospective cohort | 366/147 | 55 | 2012.6-2016.11 | 513 | 90- day mortality | 3.6 | 7 | Alcohol/HBV, HCV/Crypto-genic/NAFLD/ Autoimmune |
| Chiriac [32] | 2020 | Romania | decom-pensated cirrhosis | retrospective cohort | 49/21 | 62 | 2017.1-2017.6 | 70 | hospitalized in the ICU mortality | 5 | 8 | NA |
| Sun [35] | 2021 | China | Acute-on-chronic liver failure | retrospective cohort | 346/61 | 44.5 | 2013.1-2019.12 | 412 | 90- day mortality | 4.79 | 7 | HBV |
| Liu [36] | 2021 | China | cirrhosis | prospective cohort | 1883/700 | 51.64 | 2015.1-2019.1 | 2583 | 90- day mortality | 4.89 | 7 | HBV/Alcohol/Auto-immune/HBV and Alcohol/HBV and HEV/Others |
| Shi [41] | 2021 | China | cirrhosis | retrospective cohort | 1105/494 | 49.5 | 2014.6-2017.11 | 1599 | hepatocellular carcinoma risk | 2 | 7 | HBV |
| Tapadia [42] | 2021 | Indian | cirrhosis | prospective cohort | 180/28 | 51.5 | 2017.10-2018.11 | 208 | 30- day mortality | 5.86 | 6 | Alcohol/HBV/HCV/ NASH/Cryptogenic/ Others (AIH, PBC, PSC, Wilson's)/ Combination |
| Giabicani [43] | 2021 | France | cirrhosis | retrospective cohort | 83/33 | 57 | 2010.1-2017.6 | 116 | 28- day mortality | 0.624 | 7 | Alcohol/HBV, HCV/ NASH/other |
| Magalhães [28] | 2021 | Portugal | decom-pensated cirrhosis | retrospective cohort | 105/34 | 56 | 2010.1-2016.8 | 139 | infection risk | 3.6 | 8 | Alcoholic |

*(Continued)*

**Table 2.** (Continued)

| Author | Year | Country | Population | Study design | Male/ Female | Mean age (yrs) | Duration | Patients NO | Outcome Measure | Cut- off value | Quality score | Etiology |
|---|---|---|---|---|---|---|---|---|---|---|---|---|
| Janka [34] | 2023 | Hungary | decompensated cirrhosis | prospective cohort | 145/88 | 57.9 | 2016.12-2020.9 | 233 | 90- day mortality | 3.36 | 8 | Alcoholic/Viral/ Autoimmune/others |
| Zhang [37] | 2023 | China | cirrhosis | retrospective cohort | 283/141 | 59.9 | 2018.1-2022.12 | 424 | 30-, 90- and 180-day readmissions | NA | 7 | NA |

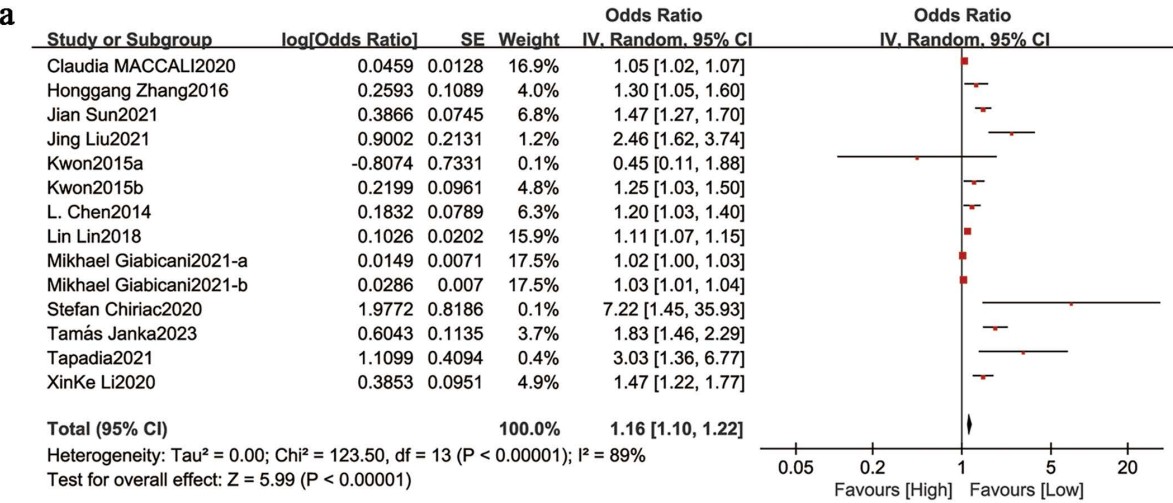

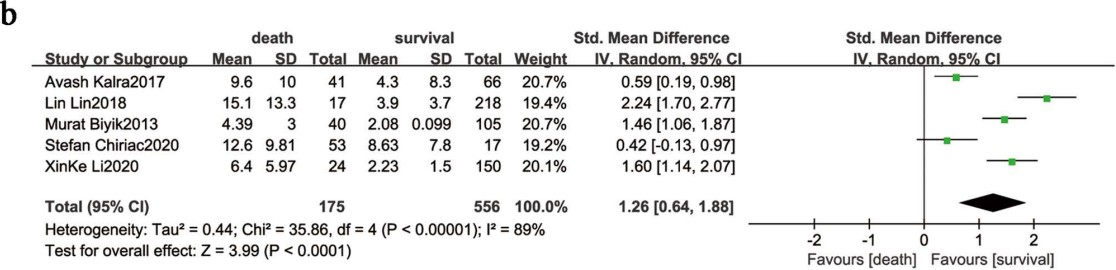

**Fig 2. (a)** Forest plots for the association between NLR and Mortality; **(b)** forest plot for the association between NIR in non-survivors versus survivors. The forest plot presents the overall results of the preliminary studies. OR (odds ratio), SE/SD (standard error/standard deviation), and NLR (neutrophil/lymphocyte ratio) are represented by different symbols: red and green squares indicate the point estimates of the odds ratios for each study, with horizontal bars showing the 95% confidence intervals; the black diamond represents the combined estimates of the overall studies along with their 95% confidence intervals.

meta-analysis revealed a statistically significant positive correlation between elevated NLR and infection (OR = 1.57, 95% CI: 1.25–1.99; P = 0.0001; Fig 3A).

Three studies provided comparative data on NLR levels between infected and non-infected cirrhotic patients. Analysis using a random-effects model revealed moderate heterogeneity ($I^2$ = 65%, P = 0.06). Pooled results demonstrated significantly elevated NLR levels in infected patients compared to uninfected controls (SMD = 0.67, 95% CI: 0.37–0.96; P < 0.0001; Fig 3B).

**Table 3. Subgroup analysis of mortality according to patient age, study design, NLR cut-off value, follow-up duration and geographic region.**

| Subgroup | Mortality | | | |
|---|---|---|---|---|
| | Study | OR [95%CI] | *P* value | $I^2$ |
| **Total** | 14 | 1.16 [1.10–1.22] | <0.00001 | 89% |
| **Study design** | | | | |
| Prospective | 5 | 1.57 [1.17–2.11] | 0.003 | 92% |
| Retrospective | 9 | 1.12 [1.07–1.18] | <0.00001 | 89% |
| **Follow-up** | | | | |
| ≥ 12 months | 3 | 1.10 [0.93–1.31] | 0.27 | 56% |
| < 12 months | 10 | 1.19 [1.12–1.26] | <0.00001 | 92% |
| **Region** | | | | |
| Asia | 9 | 1.36[1.18–1.58] | <0.0001 | 81% |
| Europe | 5 | 1.05 [1.01–1.09] | 0.02 | 89% |
| **NLR cut-off** | | | | |
| >3 | 9 | 1.52[1.24–1.85] | <0.0001 | 91% |
| ≤3 | 2 | 1.02[1.01–1.04] | 0.001 | 47% |
| **Average age** | | | | |
| >60 | 2 | 2.37 [0.39–14.36] | 0.35 | 81% |
| ≤60 | 12 | 1.16 [1.10–1.22] | <0.00001 | 90% |

**3.3.3. NLR and ascites.** Four studies evaluated NLR levels in cirrhotic patients with ascites. The random-effects model analysis showed moderate heterogeneity ($I^2 = 53\%$, P = 0.09). A statistically significant association was found between elevated NLR and ascites presence (OR = 2.03, 95% CI: 1.43–2.88; P < 0.0001; Fig 4A).

**3.3.4. NLR and hepatic encephalopathy.** Three studies assessed NLR levels in cirrhotic patients with hepatic encephalopathy. The random-effects model analysis showed minimal heterogeneity ($I^2 = 6\%$, P = 0.34). No significant association was observed between NLR and hepatic encephalopathy (OR = 1.33, 95% CI: 0.89–2.00; P = 0.16; Fig 4B).

### 3.4. Sensitivity analysis

Sensitivity analyses were conducted to assess the robustness of NLR's clinical significance. Iterative exclusion of individual studies showed the associations with mortality, infection rates, and hepatic encephalopathy incidence remained stable within the original confidence intervals. This consistency indicate that no single study unduly influenced the primary outcomes for mortality (Fig 5A–B), infection rates (Fig 5C–D), hepatic encephalopathy (S1A Fig). However, inconsistent results emerged for ascites.Exclusion of studies by Liu et al. and Ke et al. altered the statistical significance of the association, suggesting insufficient evidence to establish NLR as a reliable predictor for ascites development in cirrhosis (S1C Fig).

### 3.5. Publication bias

We assessed potential publication bias using funnel plot visualization and Egger's regression test. Funnel plot asymmetry suggested significant bias in mortality analysis (Egger's test: p = 0.0001; Fig 6A). In contrast, no statistically significant publication bias was detected for survival outcomes (Egger's test: p = 0.734; Fig 6B), infection rates (Egger's test: p = 0.087; S2A Fig), infection status comparisons (Egger's test: p = 0.730; S2B Fig), ascites subgroups (Egger's test: p = 0.912; S3A Fig), hepatic encephalopathy cohorts (Egger's test: p = 0.180; S3B Fig).

a

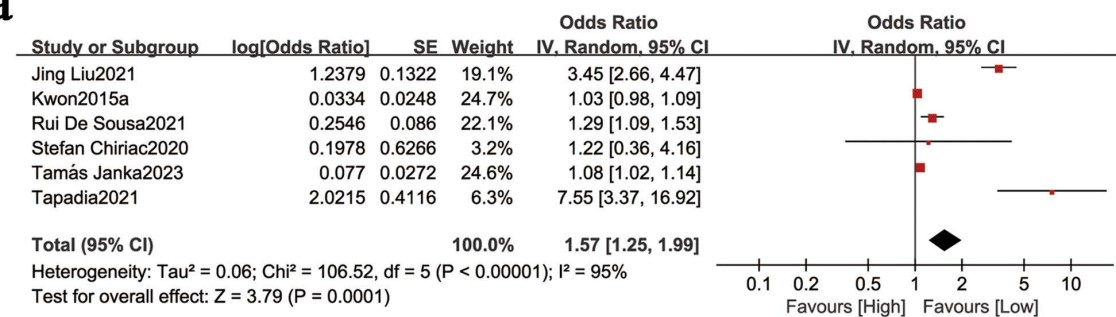

b

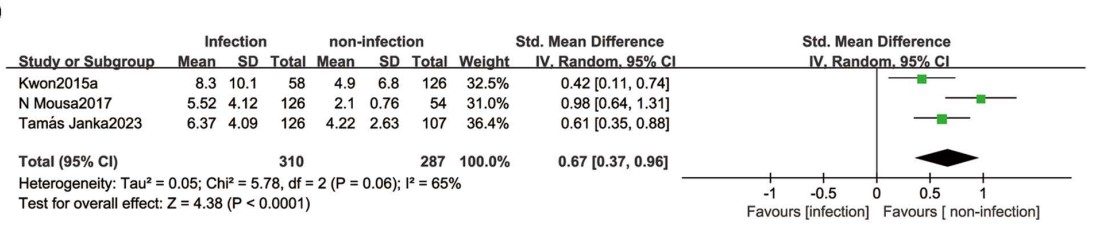

**Fig 3. (a)** Forest plot for the association between NLR and infection, **(b)** forest plot for the association between NLR in infected versus non-infected patients.The forest plot presents the overall results of the preliminary studies. OR (odds ratio), SE/SD (standard error/standard deviation), and NLR (neutrophil/lymphocyte ratio) are represented by different symbols: red and green squares indicate the point estimates of the odds ratios for each study, with horizontal bars showing the 95% confidence intervals; the black diamond represents the combined estimates of the overall studies along with their 95% confidence intervals.

## 4. Discussion

Cirrhosis progresses from a compensated phase with a favorable prognosis to an advanced phase marked by complications of portal hypertension and/or hepatic insufficiency [47,48]. This condition often causes immune system dysfunction, impairing both innate and adaptive responses and leading to systemic inflammation and immunodeficiency. The neutrophil-to-lymphocyte ratio (NLR) reflects systemic inflammation, with neutrophils representing inflammatory activity and lymphocytes indicating immune regulation [33]. NLR may thus serve as a prognostic factor by balancing these two components [49]. Its utility has been widely explored in diverse diseases [50]. In cirrhotic patients, elevated NLR is linked to systemic inflammation, which exacerbates disease progression [47,51]. Recent evidence underscores NLR as a key prognostic marker; for instance, Jing Liu et al. [36] reported that NLR levels predicted 90-day mortality, while Claudia Maccali et al. [44] identified NLR as a short-term mortality indicator in acute decompensation cases.

In our meta-analysis, we rigorously screened studies that reported odds ratios with 95% confidence intervals. A total of 18 studies comprising 7,714 patients with liver cirrhosis were included. Preliminary analysis revealed a statistically significant association between elevated NLR levels and adverse prognosis. NLR serves as a critical biomarker in cirrhosis, strongly linked to poor outcomes such as mortality and disease complications. Specifically, higher NLR values correlate with increased incidence of liver-related adverse events and escalating mortality risk. Wu et al. [52] investigated NLR's predictive utility for emergency liver transplantation in HBV-associated ACLF. Their findings demonstrated that patients with NLR > 6 had the highest mortality rates and most urgently required transplantation. These results indicate that NLR exhibits superior predictive accuracy for morbidity and mortality in liver disease, potentially serving as a practical tool to prioritize candidates for urgent transplantation.

Twelve studies reported a significant association between elevated NLR and mortality in patients with cirrhosis. Sensitivity analyses demonstrated consistent results across all models. Publication bias was evaluated through enhanced

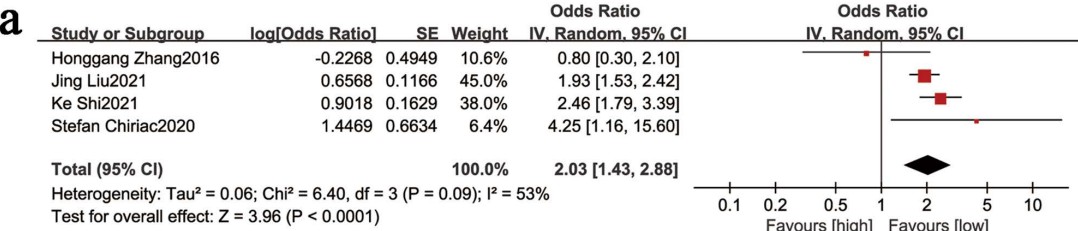

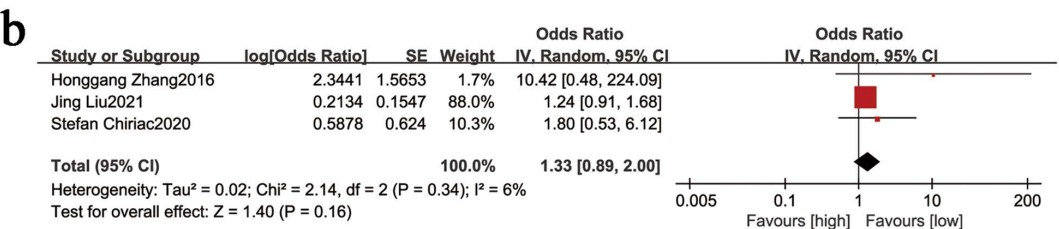

**Fig 4.** **(a)** Forest plot for the association between NLR values and ascites, **(b)** forest plot for the association between NLR and hepatic encephalopathy. The forest plot shows the combined results of the preliminary studies. OR (odds ratio), SE (standard error), and NLR (neutrophil-to-lymphocyte ratio) are represented by different symbols: red squares indicate the point estimates of the odds ratios for each study, with horizontal lines representing the 95% confidence intervals; black diamonds represent the combined estimates across studies and their 95% confidence intervals.

contour funnel plots and Egger's regression test. The funnel plot showed asymmetry, and Egger's test confirmed significant bias (p < 0.05), aligning with prior findings by Honggang Zhang et al. [31]. Elevated NLR levels are strongly linked to severe infection and systemic inflammation, as evidenced by six studies identifying NLR as a predictor of infection risk in cirrhotic individuals. Additional sensitivity analyses confirmed the robustness of these results. Notably, our assessment of publication bias revealed no significant statistical evidence (p = 0.18), consistent with the conclusions of Jung Hyun Kwon et al. [33].

Subgroup analyses were performed to explore potential sources of heterogeneity. The results indicate that variability in NLR cutoff values may account for the observed heterogeneity; therefore, future studies should aim to standardize NLR thresholds at <3. Notably, NLR demonstrates greater prognostic accuracy for short-term mortality, particularly in patients under 60 years of age.

The heterogeneity observed in this study may stem from several factors: first, significant differences in NLR cutoff values used across studies (e.g., > 3 vs. ≤ 3) directly affect the consistency of prognostic criteria; second, differences in study design (prospective vs. retrospective) and population characteristics (age, etiology) may lead to heterogeneous expression of immune-inflammatory responses; additionally, varying follow-up times (short-term vs. long-term prognosis) further increased outcome variability. These heterogeneities may impair the accuracy of NLR prediction potency. To reduce heterogeneous in future studies, we recommended: (1) determining optimal NLR cutoff values for etiology and staging specificity through multicenter studies; (2) adopting standardized study designs (e.g., unified prospective cohorts); (3) conducting mechanistic studies to elucidate the association between dynamic changes in NLR and immune-inflammatory pathways, thereby improving the clinical applicability and comparability of this market.

Liver cirrhosis (LC) is characterized by systemic inflammation and immunodeficiency [53], which are interdependent. Inflammation is marked by elevated pro-inflammatory cytokines and their circulating levels, while immune deficiency arises from hepatic damage and dysregulated systemic immune responses. These dynamic changes may underlie the diverse clinical manifestations of cirrhosis. Although the mechanism linking elevated NLR to poor outcomes in cirrhosis is not fully elucidated, a leading hypothesis implicates neutrophilia coupled with lymphocytopenia due to apoptosis [54].

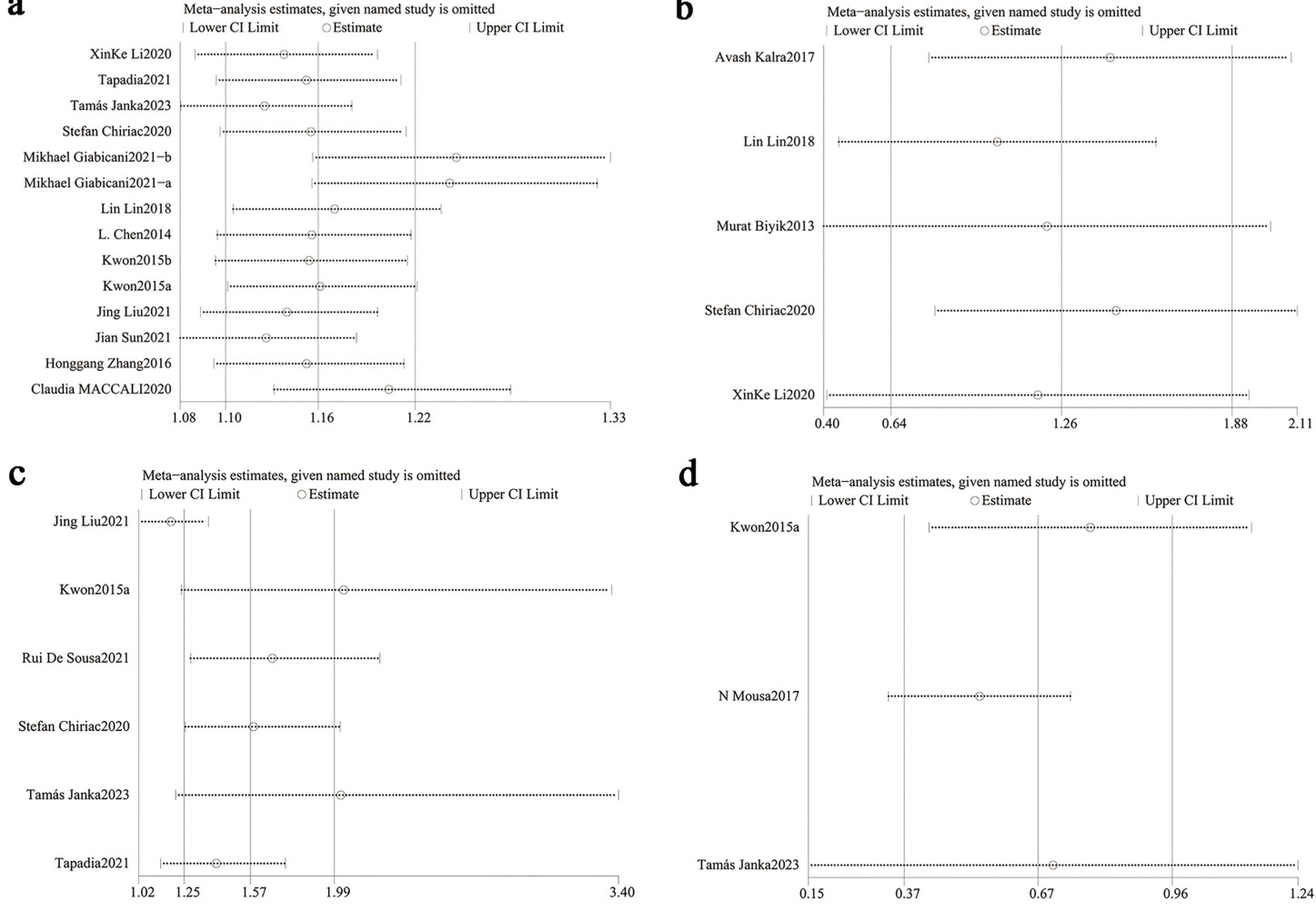

**Fig 5. (a)** Sensitivity analysis of NLR and mortality; **(b)** sensitivity analysis of non-survivors compared to survivors; **(c)** sensitivity analysis of NLR and infection; **(d)** sensitivity analysis of infected versus non-infected patients. As shown in the figure: by excluding individual studies one by one, the association between mortality and infection rates remains stable within the original confidence intervals. This consistency indicates that no single study has a significant impact on the primary outcome measures of mortality (Fig 5A and 5B) and infection rates (Fig 5C and 5D).

This state of secondary immunodeficiency compromises host defenses and increases mortality risk. Supporting this, a single-center retrospective study of 91 biopsy-proven NAFLD patients demonstrated that NLR significantly correlated with advanced fibrosis and inflammatory activity [55], confirming its role as an independent biomarker for disease progression. Secondly, patients with cirrhosis spontaneously develop a stronger pro-inflammatory response due to an imbalance of pro-inflammatory (enhancing) and anti-inflammatory (inhibitory) signaling pathways in immune cells [56–58], and this pro-inflammatory state may be associated with inhibited neutrophil apoptosis [59] and increased apoptosis of lymphocytes in the thymus and spleen [60]. Studies have shown that this neutrophil dysfunction predicts the prognosis of cirrhosis [61], possibly related to bacterial phagocytosis by neutrophils [62], Neutrophils also inhibit T cell activation by producing arginase, nitric oxide, and other substances [63], leading to lymphocyte-mediated depletion of immune responses. Therefore, an elevated neutrophil-to-lymphocyte ratio (NLR) may be a marker of poor prognosis in patients with cirrhosis.

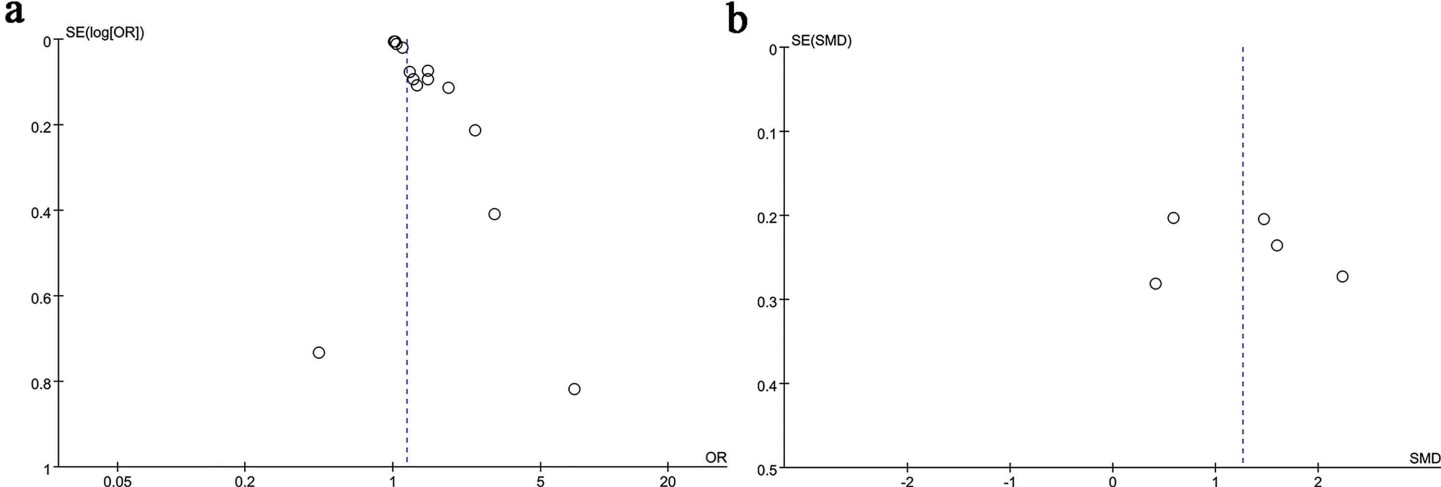

**Fig 6. (a)** Funnel plot for the evaluation of publication bias for mortality. As shown in the figure, the research points are relatively concentrated on the right side of the combined effect size, while there is a gap on the left side, showing obvious asymmetry, indicating the presence of publication bias. **(b)** Funnel plot for the evaluation of publication bias for survival and non-survival. As shown in the figure, the research points are roughly symmetrically distributed on both sides of the combined effect size, indicating that the likelihood of publication bias in this study is low.

## 5. Limitations

Our meta-analysis has several limitations: First, despite the use of random-effects models, moderate heterogeneity persists, which may affect the reliability of the conclusions. Future studies should be improved by: (1) using standardized NLR cutoffs (e.g., determining optimal cutoff values for different stages of cirrhosis through multicenter studies); (2) conducting stratified analysis by etiology (viral/alcoholic/NAFLD) and clinical stage. Second, potential publication bias exists; it is recommended that follow-up studies pre-register protocols on platforms such as PROSPERO and encourage the publication of negative results. Third, NLR, as a dynamic biomarker, may be influenced by drug treatments, future studies should record patients' medication status (e.g., antibiotics, immunomodulator). In addition, elevated NLR may reflect different hematological patterns (neutrophilia/lymphopenia/both), and larger sample sizes are needed to differentiate between these subtypes. Finally, evidence is insufficient regarding the association between NLR and the risk of readmission and liver cancer occurrence, speciallized studies are recommended to evaluate the predictive value of NLR for these outcomes. These improvements will help more accurately establish the clinical value of NLR in the prognostic assessment of liver cirrhosis.

## 6. Conclusion

NLR serves as a critical prognostic biomarker in adult liver cirrhosis, where elevated values consistently correlate with adverse clinical outcomes. Nevertheless, the pathophysiological mechanisms underlying this association require further elucidation. To validate these observations and establish NLR as a robust prognostic indicator, large-scale multicenter prospective cohort studies should be conducted to further verify its clinical value by standardizing and dynamically monitoring the association between NLR and prognosis.

## Supporting information

**S1 Fig. (a). Sensitivity analysis of hepatic encephalopathy incidence; (b) Raw data for sensitivity analysis of the incidence of hepatic encephalopathy; (c). Sensitivity analysis of NLR and ascites; (d) Raw data for sensitivity analysis of NLR and ascites.**
(TIF)

**S2 Fig. (a). Funnel plot for publication bias for infection; (b). Funnel plot for publication bias for infected vs. non-infected patients.**

(TIF)

**S3 Fig. (a). Funnel plot for publication bias for ascites; (b). Funnel plot for publication bias for hepatic encephalopathy.**

(TIF)

**S1 Appendix. PRISMA checklist.**

(DOCX)

**S2 Appendix. Minimum dataset.**

(XLS)

## Author contributions

**Methodology:** Jingjing Lin, Mengna Huang.

**Supervision:** Lina Shen.

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
