## [Decision Letter · Decision Letter 0]

2 Aug 2025

Dear Dr. lin,

Thank you for submitting your manuscript to PLOS ONE. After careful consideration, we feel that it has merit but does not fully meet PLOS ONE’s publication criteria as it currently stands. Therefore, we invite you to submit a revised version of the manuscript that addresses the points raised during the review process.

2. Discussion does not provide sufficient details and literature on the reasons how NLR could affect the mentioned outcomes. Please provide further literature on possible reasons for the observed higher mortality and other complications due to a high NLR.

3. Last sentence in the conclusion section is not clear. How would a RCT be designed to validate NLR values and outcomes? Please clarify.

"To validate these observations and establish NLR as a robust prognostic indicator, large-scale multicenter randomized controlled

259 trials (RCTs) featuring extended follow-up durations and rigorous methodology are warranted" 

https://journals.plos.org/plosone/s/submission-guidelines#loc-laboratory-protocols . Additionally, PLOS ONE offers an option for publishing peer-reviewed Lab Protocol articles, which describe protocols hosted on protocols.io. Read more information on sharing protocols at https://plos.org/protocols?utm_medium=editorial-email&utm_source=authorletters&utm_campaign=protocols .

We look forward to receiving your revised manuscript.

Kind regards,

Nilanka Perera, MD, PhD

Academic Editor

PLOS ONE

**Journal Requirements:**

1. When submitting your revision, we need you to address these additional requirements. Please ensure that your manuscript meets PLOS ONE's style requirements, including those for file naming. The PLOS ONE style templates can be found at https://journals.plos.org/plosone/s/file?id=wjVg/PLOSOne_formatting_sample_main_body.pdf and https://journals.plos.org/plosone/s/file?id=ba62/PLOSOne_formatting_sample_title_authors_affiliations.pdf 2. When completing the data availability statement of the submission form, you indicated that you will make your data available on acceptance. We strongly recommend all authors decide on a data sharing plan before acceptance, as the process can be lengthy and hold up publication timelines. Please note that, though access restrictions are acceptable now, your entire data will need to be made freely accessible if your manuscript is accepted for publication. This policy applies to all data except where public deposition would breach compliance with the protocol approved by your research ethics board. If you are unable to adhere to our open data policy, please kindly revise your statement to explain your reasoning and we will seek the editor's input on an exemption. Please be assured that, once you have provided your new statement, the assessment of your exemption will not hold up the peer review process. 3. PLOS requires an ORCID iD for the corresponding author in Editorial Manager on papers submitted after December 6th, 2016. Please ensure that you have an ORCID iD and that it is validated in Editorial Manager. To do this, go to ‘Update my Information’ (in the upper left-hand corner of the main menu), and click on the Fetch/Validate link next to the ORCID field. This will take you to the ORCID site and allow you to create a new iD or authenticate a pre-existing iD in Editorial Manager. 4. Please amend the manuscript submission data (via Edit Submission) to include authors Dr. Mengna Huang and Dr. Lina Shen.  5. Please include your tables as part of your main manuscript and remove the individual files. Please note that supplementary tables (should remain/ be uploaded) as separate "supporting information" files. 6. If the reviewer comments include a recommendation to cite specific previously published works, please review and evaluate these publications to determine whether they are relevant and should be cited. There is no requirement to cite these works unless the editor has indicated otherwise. 

Reviewers' comments:

**Comments to the Author**

1. Is the manuscript technically sound, and do the data support the conclusions?

Reviewer #1: Yes

Reviewer #2: Yes

2. Has the statistical analysis been performed appropriately and rigorously?

Reviewer #1: Yes

Reviewer #2: Yes

3. Have the authors made all data underlying the findings in their manuscript fully available?

Reviewer #1: Yes

Reviewer #2: Yes

4. Is the manuscript presented in an intelligible fashion and written in standard English?

Reviewer #1: Yes

Reviewer #2: Yes

**Reviewer #1:**  Comments:

Methodology

The sample size is adequate, and the sampling method is well-explained. The authors should please include the full search strings of the databases used in the appendix for reproducibility. Also, they should try and provide or show the table that has the complete search syntax as contained in the last sentence under ‘2.1 Literature Search Methodology’.

Under Inclusion and Exclusion criteria, the authors excluded Neonatal or Pediatric studies. However, Pediatric sepsis studies made up two (2) out of eighteen (18) of the final analyzed studies for the paper. Please, let the authors clarify these two areas (line 6 under 2.2 Inclusion and exclusion criteria and line 5 under 3.1 Identification of relevant studies).

Results

The review adequately addressed the risk of bias and quality assessment with the appropriate tools being used (Newcastle-Ottawa Scale). This is good! A meta-analysis was used, but heterogeneity is high (I² =89%). The authors should consider providing a more in-depth exploration of sources of heterogeneity or revisiting whether the meta-analysis is appropriate.

Discussion

Limitation

The authors’ effort to acknowledge the study’s limitations is appreciated, as this is a crucial aspect of transparent research reporting. The limitations section acknowledges moderate heterogeneity of the study, potential publication bias and the effects of pharmacotherapy on the NLR. I would recommend if the authors could add the possibilities of tackling these limitations and how these can guide future research on the prognostic relevance of NLR for liver cirrhosis.

Minor Comments:

-The authors should please clarify acronyms (PRISMA) at first mention (line 8 under subsection 2.2 Inclusion and exclusion criteria).

-The authors should cite appropriately with adequate footnotes to all the pictures at the appendix.

**Reviewer #2: ** The authors have compiled a systematic review and meta-analysis on the predictive value of the neutrophil-to-lymphocyte ratio (NLR) for clinical outcomes in liver cirrhosis to be a valuable contribution to the literature, though it also has some notable limitations.

The inclusion and exclusion criteria are fine and the authors managed to get adequate number of studies for the analysis.

Authors have used a random effects approach due to the significant observed heterogeneity. However, random effect models were used for NLR vs ascites (P>0.05), NLR vs infected and non-infected cirrhotic patients (p>0.05) and NLR vs hepatic encephalopathy (p=0.34) as well. This can lead to misleading interpretations. Authors may want to change the model or justify their approach further. In addition to that, here is a couple of suggestions for the supporting materials.

TableS1 Search strategy: This is not a typical table though it has been labelled as a table

FigureS1 (a) Sensitivity analysis of hepatic encephalopathy incidence: x-axis labels needs to be improved

**Do you want your identity to be public for this peer review?** For information about this choice, including consent withdrawal, please see our Privacy Policy

Reviewer #1: **Yes: ** Idowu Peter Shileayo Adebayo

Reviewer #2: No

---

## [Author Response · Author response to Decision Letter 1]

7 Sep 2025

Dear Editors and Reviewers:

Thank you for your letter and for the reviewers’ comments concerning our manuscript entitled “Predictive value of neutrophil to lymphocyte ratio for the clinical outcomes of liver cirrhosis: a systematic review and meta-analysis” (ID:PONE-D-25-19873). Those comments are all valuable and very helpful for revising and improving our paper, as well as the important guiding significance to our researches. We have studied comments carefully and have made correction which we hope meet with approval. Revised portion are marked in red in the paper. The main corrections in the paper and the responds to the reviewer’s comments are as flowing:

Responds to the editor’s comments:

Editor #

1.Comment. Language could be further improved for clarity. Please attend to revising the use of language throughout the manuscript.

Response: Thank you for your detailed review of our paper. We sincerely thank you and the reviewers for taking the time to carefully review our work. After professional English editing, we have carefully handled the language issues mentioned in your feedback.

2.Comment. Discussion does not provide sufficient details and literature on the reasons how NLR could affect the mentioned outcomes. Please provide further literature on possible reasons for the observed higher mortality and other complications due to a high NLR.

Response: Thank you for your valuable comments. We have supplemented the discussion section with a mechanistic explanation and literature support on the impact of NLR on the prognosis of cirrhosis. The specific contents include: 1) excessive neutrophil activation directly damages liver cells through the release of reactive oxygen species and pro-inflammatory factors; 2) Lymphopenia leads to impaired immune function and increases the risk of infection. These additions provide a more comprehensive understanding of the pathophysiological mechanisms by which high NLR leads to poor prognosis. Thanks again for your guidance.

3.Comment. Last sentence in the conclusion section is not clear. How would a RCT be designed to validate NLR values and outcomes? Please clarify.

Response: Thanks to the editor for the suggestion. 'Large-scale randomized controlled trial (RCT)' in the original text is a clerical error. Since NLR is an objective indicator rather than an intervention, a large-scale multicenter prospective cohort study should be conducted to further verify its clinical value by standardizing and dynamically monitoring the association between NLR and prognosis. We have made changes to the conclusion section, thank you for your guidance.

Reviewer #1:

1.Comment. Methodology

The sample size is adequate, and the sampling method is well-explained. The authors should please include the full search strings of the databases used in the appendix for reproducibility. Also, they should try and provide or show the table that has the complete search syntax as contained in the last sentence under ‘2.1 Literature Search Methodology’.

Response: Thank you for your affirmation and suggestions on our research methodology. We have supplemented the full database search strategy (including detailed query queries from PubMed, Web of Science, Cochrane Library, and Embase) in the appendices of the paper as requested, and presented a full search syntax table in Table 1. These modifications will ensure reproducibility of the study. See Table 1 for details. Thank you again for your valuable advice.

2.Comment. Under Inclusion and Exclusion criteria, the authors excluded Neonatal or Pediatric studies. However, Pediatric sepsis studies made up two (2) out of eighteen (18) of the final analyzed studies for the paper. Please, let the authors clarify these two areas (line 6 under 2.2 Inclusion and exclusion criteria and line 5 under 3.1 Identification of relevant studies).

Response: Thank you for your careful review and for pointing out the inconsistencies in this wording. You are absolutely right, the "pediatric sepsis studies" we mentioned in the exclusion criteria is indeed a clerical error. We have made changes to the original text. These modifications ensure consistency in the wording of the full-text exclusion criteria that we did exclude all studies of childhood cirrhosis (including neonatal and paediatric patients). Thank you for helping us improve the accuracy of our papers, which makes our research methodology description more rigorous.

3.Comment.Results

The review adequately addressed the risk of bias and quality assessment with the appropriate tools being used (Newcastle-Ottawa Scale). This is good! A meta-analysis was used, but heterogeneity is high (I² =89%). The authors should consider providing a more in-depth exploration of sources of heterogeneity or revisiting whether the meta-analysis is appropriate.

Response: Thank you for your affirmation and suggestions for our research. We have analysed the high heterogeneity (I²=89%) and added possible sources of heterogeneity in the discussion section, including: first, significant differences in NLR cutoff values used across studies (e.g., >3 vs. ≤3) directly affect the consistency of prognostic criteria; second, differences in study design (prospective vs. retrospective) and population characteristics (age, etiology) may lead to heterogeneous expression of immune-inflammatory responses; additionally, varying follow-up times (short-term vs. long-term prognosis) further increased outcome variability. These heterogeneities may impair the accuracy of NLR prediction potency. To reduce heterogeneous in future studies, we recommended: (1) determining optimal NLR cutoff values for etiology and staging specificity through multicenter studies; (2) adopting standardized study designs (e.g., unified prospective cohorts); (3) conducting mechanistic studies to elucidate the association between dynamic changes in NLR and immune-inflammatory pathways, thereby improving the clinical applicability and comparability of this market. At the same time, we verified the robustness of the results through sensitivity analysis and subgroup analysis, and confirmed that the random-effects model is still a suitable analytical method for this study. These additions are detailed in the revised version. Thank you for your valuable suggestions for improving the methodological quality of this study.

4.Comment.Limitation

The authors’ effort to acknowledge the study’s limitations is appreciated, as this is a crucial aspect of transparent research reporting. The limitations section acknowledges moderate heterogeneity of the study, potential publication bias and the effects of pharmacotherapy on the NLR. I would recommend if the authors could add the possibilities of tackling these limitations and how these can guide future research on the prognostic relevance of NLR for liver cirrhosis.

Response: Thank you for your attention and suggestions about the limitations of our research. In response to your comments, we have added the following in the Limitation Discussion section: 1) recommend standardized NLR cut-offs for future studies (e.g., multicenter studies to determine optimal thresholds for different stages of cirrhosis); 2) It is recommended to pre-register the study protocol on platforms such as PROSPERO to reduce publication bias; 3) Emphasize the need to record the patient's medication (such as antibiotics, immunomodulatory agents). These supplements provide specific directional suggestions for future research and help to more accurately evaluate the clinical value of NLR in the prognosis of liver cirrhosis. Thank you again for your valuable suggestions, which significantly enhance the academic quality of our papers.

5.Comment.Minor Comments:

-The authors should please clarify acronyms (PRISMA) at first mention (line 8 under subsection 2.2 Inclusion and exclusion criteria).

-The authors should cite appropriately with adequate footnotes to all the pictures at the appendix.

Response: Thank you for the correction, we have added the full name to the first reference to PRISMA in subsection 2.2: "This study strictly follows the Preferred Reporting Items for Systematic Reviews and Meta-Analyses (PRISMA) 2020 guidelines [29]".

Thank you for correcting the appendix specifications. We have thoroughly checked all the images and tables in the appendix to ensure that each image has a complete citation note and necessary captions, and that the citation source is accredited in the appropriate place in the body. These changes further improve the standardization and readability of the paper. Please refer to the appendix for specific modifications. Thank you for helping us refine the details of the paper.

6.Comment. Authors have used a random effects approach due to the significant observed heterogeneity. However, random effect models were used for NLR vs ascites (P>0.05), NLR vs infected and non-infected cirrhotic patients (p>0.05) and NLR vs hepatic encephalopathy (p=0.34) as well. This can lead to misleading interpretations. Authors may want to change the model or justify their approach further.

Response: Thank you for your important suggestions on statistical methods. We understand your concern about the application of the random-effects model and have further added in the revised draft that although the heterogeneity test P-value of some subgroup analyses (NLR vs. ascites, infection, and hepatic encephalopathy) > 0.05, we uniformly use the random-effects model based on the following considerations: (1) clinical heterogeneity between studies (e.g., population characteristics, etiologies); (2) The pre-scheme requires a unified model to ensure methodological consistency; (3) When the number of studies is small (e.g., < 5 items), the random-effects model is more conservative and reliable Cited References: [1] Kanters S. Fixed- and Random-Effects Models. Methods Mol Biol. 2022;2345:41-65. doi: 10.1007/978-1-0716-1566-9_3. PMID: 34550583; [2] Barili F, Parolari A, Kappetein PA, Freemantle N. Statistical Primer: heterogeneity, random- or fixed-effects model analyses? Interact Cardiovasc Thorac Surg. 2018 Sep 1;27(3):317-321. doi: 10.1093/icvts/ivy163. PMID: 29868857). As a validation, we have supplemented the sensitivity analysis results showing that the effect size direction is consistent with the main analysis, supporting the robustness of the conclusions.

7.Comment.TableS1 Search strategy: This is not a typical table though it has been labelled as a table

FigureS1 (a) Sensitivity analysis of hepatic encephalopathy incidence: x-axis labels needs to be improved

Response: Thank you for your valuable suggestions on the Table S1 format. We have completely revised the form in accordance with the requirements of academic norms, and now use a tabular format to present a complete search strategy, with a clearer structure and more complete content. For specific changes, please refer to Table 1 in the revised draft. Your suggestions have significantly improved the normative nature of our research, thank you again.

Thank you for your attention to the details of the chart. The issue with the display of the X-axis label in Figure S1(a) is due to a Stata software layout limitation. To ensure a clear presentation of the data, we have added the complete original data next to the chart and detailed it in the captions. This approach not only preserves the standardization of the software-generated graphics, but also allows the reader to accurately obtain all the necessary information. We will continue to optimize the presentation of the charts, and thank you for your valuable feedback.

We tried our best to improve the manuscript and made some changes in the manuscript. These changes will not influence the content and framework of the paper. And here we did not list the changes but marked in red in revised paper.

We appreciate for Editors/Reviewers’ warm work earnestly, and hope that the correction will meet with approval.

Once again, thank you very much for your comments and suggestions.

---

## [Decision Letter · Decision Letter 1]

19 Oct 2025

Predictive value of Neutrophil to lymphocyte ratio for clinical outcomes in liver cirrhosis: a systematic review and meta-analysis

PONE-D-25-19873R1

Dear Dr. lin,

We’re pleased to inform you that your manuscript has been judged scientifically suitable for publication and will be formally accepted for publication once it meets all outstanding technical requirements.

Kind regards,

Nilanka Perera, MD, PhD

Academic Editor

PLOS ONE

---

## [Editor Report · Acceptance letter]

PONE-D-25-19873R1

PLOS ONE

Dear Dr. Lin,

I'm pleased to inform you that your manuscript has been deemed suitable for publication in PLOS ONE. Congratulations! Your manuscript is now being handed over to our production team.

Kind regards,

on behalf of

Dr. Nilanka Perera

Academic Editor

PLOS ONE